# Shock Index: A Simple and Effective Clinical Adjunct in Predicting 60-Day Mortality in Advanced Cancer Patients at the Emergency Department

**DOI:** 10.3390/ijerph17134904

**Published:** 2020-07-07

**Authors:** Tzu-Heng Cheng, Yi-Da Sie, Kuang-Hung Hsu, Zhong Ning Leonard Goh, Cheng-Yu Chien, Hsien-Yi Chen, Chip-Jin Ng, Chih-Huang Li, Joanna Chen-Yeen Seak, Chen-Ken Seak, Yi-Tung Liu, Chen-June Seak

**Affiliations:** 1Department of Emergency Medicine, Lin-Kou Medical Center, Chang Gung Memorial Hospital, Taoyuan 33305, Taiwan; b9502086@cgmh.org.tw (T.-H.C.); hshychen@gmail.com (H.-Y.C.); ngowl@ms3.hinet.net (C.-J.N.); y17322@cgmh.org.tw (C.-H.L.); julianseak@sem.org.tw (S.I.); 2College of Medicine, Chang Gung University, Taoyuan 33302, Taiwan; 3Department of Emergency Medicine, New Taipei Municipal Tucheng Hospital, New Taipei City 23652, Taiwan; 4Department of Emergency Medicine, China Medical University Hospital, Taichung 404332, Taiwan; easythinking20@gmail.com; 5Laboratory for Epidemiology, Department of Health Care Management, and Healthy Aging Research Center, Chang Gung University, Taoyuan 33302, Taiwan; khus@mail.cgu.edu.tw; 6Sarawak General Hospital, Kuching, Sarawak 93586, Malaysia; lgzn92@gmail.com (Z.N.L.G.); joannaseak@hotmail.com (J.C.-Y.S.); jonathanseak@gmail.com (C.-K.S.); 7Department of Emergency Medicine, Ton-Yen General Hospital, Zhubei, Hsinchu County 30268, Taiwan; rainccy217@gmail.com; 8School of Medicine, National Defense Medical Center, Taipei 11490, Taiwan; tung.9026.yt@gmail.com

**Keywords:** shock index, advanced cancer, emergency physicians, emergency department, 60-day survival, Stratification to Prevent Overcrowding Taskforce (SPOT)

## Abstract

Deciding between palliative and overly aggressive therapies for advanced cancer patients who present to the emergency department (ED) with acute issues requires a prediction of their short-term survival. Various scoring systems have previously been studied in hospices or intensive care units, though they are unsuitable for use in the ED. We aim to examine the use of a shock index (SI) in predicting the 60-day survival of advanced cancer patients presenting to the ED. Identified high-risk patients and their families can then be counseled accordingly. Three hundred and five advanced cancer patients who presented to the EDs of three tertiary hospitals were recruited, and their data retrospectively analyzed. Relevant data regarding medical history and clinical presentation were extracted, and respective shock indices calculated. Multivariate logistic regression analyses were performed. Receiver operating characteristic (ROC) curves were plotted to evaluate the predictive performance of the SI. Nonsurvivors within 60 days had significantly lower body temperatures and blood pressure, as well as higher pulse rates, respiratory rates, and SI. Each 0.1 SI increment had an odds ratio of 1.39 with respect to 60-day mortality. The area under the ROC curve was 0.7511. At the optimal cut-off point of 0.94, the SI had 81.38% sensitivity and 73.11% accuracy. This makes the SI an ideal evaluation tool for rapidly predicting the 60-day mortality risk of advanced cancer patients presenting to the ED. Identified patients can be counseled accordingly, and they can be assisted in making informed decisions on the appropriate treatment goals reflective of their prognoses.

## 1. Introduction

The management of patients with underlying advanced cancer nearing the end of life is often palliative in order to allow them and their families to prepare for the eventuality of death while ensuring that the patients retain a reasonable quality of life [1]. Intensive life-sustaining therapies in these patients inflict unnecessary suffering without significant improvement in survival rates [2,3]. This applies not only to the patients’ underlying condition of advanced cancer but also for acute presentations to the emergency department (ED), which may or may not be related to the patients’ cancer diagnosis and disease progression. Early integration of palliative care requires the prediction of a patient’s short-term survival [4], and this should ideally be assessed upon presentation to the ED. Initiating palliative care right from ED presentation for patients with advanced cancer improves their quality of life without adversely affecting survival rates [5].

However, doctors—emergency physicians (EPs), internists, oncologists, and intensivists alike—do not have an objective method of rapidly assessing the short-term mortality risk of advanced cancer patients in the ED. This often results in physicians erring on the side of caution and opting to pursue aggressive treatment regardless of the patient’s survival probability, going against the ideal practice of palliation for terminally ill patients. Consequently, hospital facilities are overused while hospice care is underused [6], representing a misallocation of resources in addition to unnecessary jeopardy of the patient’s quality of life.

Various scoring systems have previously been studied in hospices or intensive care unit settings, but they are unsuitable for use in the ED. Some scoring systems require laboratory data that is not available at all ED settings [7,8,9,10,11], while others require time-consuming, complex calculations which hamper the swift decision-making process required in the ED [11,12,13]. The currently-studied scores may also incorporate subjective performance status evaluation, which can vary according to the assessor [9,10,11,13,14].

Of the limited studies done in the ED into the use of cancer disease progression and sequential organ failure assessment to predict mortality, the former requires extensive medical history while the latter requires extensive clinical and laboratory data [15,16]. Both do not enable doctors to quickly identify advanced cancer patients for whom supportive, symptomatic management and hospice care referral would be more appropriate compared to aggressive interventions.

Shock index (SI) is defined as the ratio of pulse rate to systolic blood pressure [17], both vital sign measurements which can be rapidly obtained under a minute. It has been previously studied in the prognostication of acute pulmonary embolism [18], pneumonia [19], influenza [20], trauma [21], acute myocardial infarctions [22], sepsis [23], and stroke [24]. The SI is a convenient and rapid assessment tool ideal for the evaluation of patients in the ED because of its simplicity. A normal SI is generally accepted to be within 0.5 to 0.7, while an index above 0.9 spells poor outcomes for the patient [17,25]. The SI has also been studied in the hospice setting to predict the three-day mortality risk of terminal cancer patients [25].

A period of three days is, however, hardly sufficient for patients and families to carefully deliberate over tough end-of-life matters. Furthermore, risk stratification performed in the hospice setting would have limited effectiveness in increasing the uptake of palliative care and avoiding futile aggressive interventions in the ED. We are of the opinion that such stratification can be done much earlier in advanced cancer patients, even before arriving at the hospice, to maximize its utility.

Llobera et al. found that median duration from the inception of the terminal period to death was 59 days [26], while Steensma and Loprinzi also noted that median survival of patients enrolled in Mayo Clinic’s hospice program was exactly 59 days; there were however a significant number of patients who were enrolled for less than a week [27]. With this in mind, if patients are unlikely to survive more than 59 days from the point of presentation in the ED, their families should be notified and counseled on the possible futility of invasive and aggressive treatment. This would allow them and their families to discuss and make as informed a decision as possible on how to proceed with the patients’ medical care, as well as to have sufficient time to put any outstanding matters in order.

In this current study, we studied the use of the SI as a simple risk stratification tool in the ED setting to predict 60-day mortality of advanced cancer patients in hopes of identifying candidates who will benefit from early referral to hospices. Such early inclusion of palliative services would allow for a more thorough evaluation by palliative medicine specialists and closer communication with patients and their caregivers in the ED, thus avoiding unnecessary invasive medical or surgical interventions.

## 2. Materials and Methods

### 2.1. Study Design

This retrospective study was conducted at the EDs of three training and research hospitals—Linkou Chang Gung Memorial Hospital (3406 beds with approximately 15,000 ED visits monthly in 2019), Keelung Chang Gung Memorial Hospital (1089 beds with approximately 5700 ED visits monthly in 2019), and Taipei Chang Gung Memorial Hospital (252 beds with approximately 4200 ED visits monthly in 2019). This study was approved by the Chang Gung Medical Foundation Institutional Review Board (IRB No. 104-4821B), waiving the need for patient consent. All data were accessed anonymously.

### 2.2. Settings and Subjects

We recruited all adult advanced cancer patients above the age of 18 years who visited the EDs of the three hospitals from Jan 2018 to June 2018. Advanced cancer is defined as locally recurrent or metastatic solid cancer that is not amenable to curative treatment [28,29]. Patients lost to follow-up after hospital discharge were excluded from this study, as were those who presented with out-of-hospital cardiac arrests or trauma. All patients received appropriate initial treatments in accordance with our hospital’s standard treatment protocol approved by the ED committee, based on their history of presenting illness, initial clinical evaluation, and vital signs upon presentation.

### 2.3. Measurement of Variables

Hospital records of all recruited patients were examined by trained hospital personnel to extract pertinent details regarding the patients’ medical history and vital signs at the EDs. The SI is defined as the ratio of pulse rate to systolic blood pressure calculated by dividing the pulse rate by systolic blood pressure. These calculations were performed by a general practitioner blinded to the study objectives and patient outcomes. Our primary outcome was short-term survival, defined as survival of 60 days after ED admission. The study endpoint was taken at 60 days post-ED presentation or in-hospital mortality.

### 2.4. Statistical Analysis

The means ± SD of numerical variables and frequencies with corresponding percentages of categorical variables were analyzed with univariate analyses of independent sample *t*-tests and χ2 tests, respectively. Univariate and multivariate logistic regression analyses were subsequently employed to evaluate the odds ratio of clinical parameters and each SI increment of 0.1 with respect to 60-day mortality. A receiver operating characteristic (ROC) curve was also plotted to examine the predictive performance of the SI, with the optimal cut-off point identified via maximizing Youden’s index. Kaplan–Meier analysis was also performed to examine survival between groups with high versus low SIs. Statistical significance was taken at *p* < 0.05.

## 3. Results

Within the study period of Jan 2018 to June 2018, 350 adult advanced cancer patients who visited our EDs fulfilled the inclusion criteria. Forty-five patients were subsequently excluded due to trauma (*n* = 16), out-of-hospital cardiac arrest (*n* = 10), and loss to follow-up (*n* = 19), making up a final total study population of 305 patients. Comparison of medical history between both groups of survivors versus nonsurvivors found no significant differences in terms of age, gender, type of cancer, previous treatment history, and underlying comorbidities (Table 1).

Among the 305 patients who were enrolled, 188 patients died within 60 days, with the mean survival time of 16.78 days. With respect to differences between the clinical presentations of both groups, univariate analyses found the following significant findings: nonsurvivors had lower body temperatures, higher pulse rates, higher respiratory rates, lower systolic and mean arterial blood pressure compared to survivors. The SI also differed significantly between both groups—the mean SI for survivors versus nonsurvivors was 0.92 versus 1.21, respectively (Table 2).

These variables (body temperatures, pulse rates, respiratory rates, systolic and mean arterial blood pressure, shock index) further underwent a backward model selection process using multiple logistic regression analysis. We found that the SI was the only factor that was significantly related to short-term survival in the multiple logistic regression analysis. After adjusting for age, gender, personal medical and medication history, the odds of death at 60 days after the index visit was 1.39 (95% CI: 1.24–1.55; *p* < 0.0001) times higher among participants with each 0.1 increment of the index. The area under the ROC curve was found to be 0.7511 (Hosmer–Lemeshow statistic *p*-value 0.2878; Figure 1), while Harrell’s C-index was 0.74.

At an optimal cut-off point of 0.94 (Youden’s index 0.412), the SI had a sensitivity of 81.38% in predicting 60-day mortality of advanced cancer patients (Table 3).

The Kaplan–Meier curve reveals that the 60-day mortality in advanced cancer patients with SI > 0.94 is significantly higher than those with lower SI (Wilcoxon test *p* < 0.0001; Figure 2).

## 4. Discussion

In this study, we found that there were no significant differences between the medical histories of advanced cancer patients who survived and those who did not. That a snapshot of the patient’s characteristics is a poor indicator of survival rates is perhaps a huge contributory reason behind why physicians, patients, and their families often overestimate life expectancy [27,30,31]. Even with the highly-detailed records kept by a cancer center of a patient’s cancer progression and treatment history, Geraci et al. found that disease progression predicted 180-day mortality accurately only 75% of the time [15].

Logistic regression analyses of clinical parameters on the other hand found that nonsurvivors had significantly different vital signs when compared to survivors. This finding suggests that basic clinical evaluation such as vital signs can be utilized to rapidly identify advanced cancer patients who may be better managed palliatively in a hospice rather than an acute care setting. Indeed, our further study into the use of an SI revealed that at an optimal cut-off point of 0.94, it had the high sensitivity of 81.38% in predicting 60-day mortality. Its corresponding accuracy of 73.11% also parallels that of cancer progression in the aforementioned study by Geraci et al.

This suitability of an SI in predicting 60-day mortality of advanced cancer patients in the ED is most likely because of its relationship to performance status of the circulatory system. High shock indices suggest hypovolemic or normovolemic circulatory failure [32], which often plays a part in the demise of advanced cancer patients. Advanced cancer leads to circulatory failure both directly and indirectly. Generalized cachexia, a pathognomonic sign of cancer, accounts for more than 20% of cancer mortality on its own [33], while the accompanying cardiac cachexia causes decreased left ventricular function and eventual heart failure [34,35,36,37]. This is compounded by elevated levels of circulating cardiac biomarkers, which suggest subclinical functional and morphological myocardial damage secondary to cancer progression [38]. Advanced cancer also leads to circulatory compromise indirectly through its other pathognomonic symptom of anorexia, which leads to poor nutrition and dehydration. It ultimately results in cytokine release and inflammation, causing further deterioration of cardiac function [31]. These adverse effects on the functional status of the circulatory system are then reflected as a pathological elevation of the advanced cancer patient’s SI.

The SI lends itself to effective use in the ED not just because of its ability to be rapidly calculated and interpreted, but also due to its simplicity. This would allow even junior EPs and doctors (house officers or interns), who are usually the first-responders, to initiate the discussion of prognosis and advance care planning right from the beginning. Counselling identified patients and their families regarding end-of-life issues and care can be especially important at the ED, given that patients generally put off such planning with their oncologists till they are “seriously ill”—ill enough to warrant an urgent trip to the ED; additionally, these patients have been found to be more willing to discuss end-of-life care with a house officer over their own oncologist [39]. After this topic has been broached by the first-responders or EPs, oncologists and/or the patients’ primary care doctors would then be able to continue and maintain an ongoing discussion with all pertinent stakeholders to optimize subsequent care in line with the wishes of the patients and their families.

The use of an SI to predict 60-day mortality of advanced cancer patients in the ED will inevitably change the treatment decisions of some, if not most, cancer patients. Weeks et al. found that patients who expected not to live past 6 months opted for comfort care over aggressive, life-extending therapy, while the converse was true for those who expected to live 6 months or more. Both groups, however, had similar 6-month survival rates [31]. The SI would thus be invaluable in the ED to provide patients with a better perception of their prognoses and allow them to adjust their expectations accordingly. Identified patients can then thoroughly consider their preferred treatment plan—between hospice care to preserve quality of life, or aggressive treatment which could potentially be futile. Furthermore, the implementation of this risk stratification at the ED before therapy is initiated assists families and physicians in making advance decisions regarding do-not-resuscitate status and end-of-life care, minimizing the ethical dilemma of whether to withhold or withdraw treatment in the event of the patients’ clinical deterioration. Last but not least, it must be reiterated that the SI provides an additional point of consideration in the complex decision-making process of end-of-life medical care plans at the ED; it is not to be construed as the sole definitive factor in deciding to transfer the patient to palliative care. High shock indices should prompt referrals to palliative teams at the ED for more comprehensive evaluation and discussion about further treatment plans.

While our study model has been adjusted for patients’ medications and concurrent diseases, future applications of the shock index may need to consider these variables’ effects on hemodynamic status and vital sign readings. The results of this study should be prospectively validated with a larger sample population of advanced cancer patients, and ideally with different subsets of patients with different advanced cancers. This would help to ensure its replicability across all types of cancer. Further studies into other scoring systems suitable for the ED setting may also be considered.

## 5. Conclusions

The SI is an ideal evaluation tool for rapidly predicting the 60-day mortality risk of advanced cancer patients presenting to the ED with acute issues. Its simplicity allows for all doctors, even junior EPs, to be empowered to initiate discussions regarding end-of-life issues with identified patients and their families. These patients will subsequently be able to make informed decisions in the ED between palliative care and life-extending therapy based on accurate perceptions of their prognoses. After this sensitive topic has been broached by the first-responding doctor, the discussions with all pertinent stakeholders can then be continued and coordinated to optimize treatment in line with their wishes.

## Figures and Tables

**Figure 1 ijerph-17-04904-f001:**
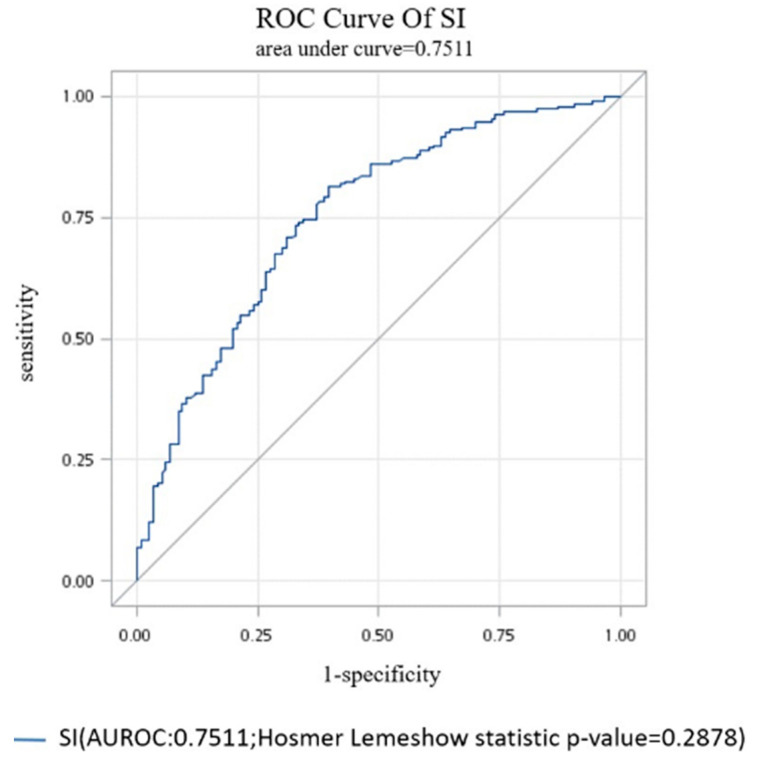
Receiver operating characteristic curve of the shock index (SI) for predicting 60-day mortality.

**Figure 2 ijerph-17-04904-f002:**
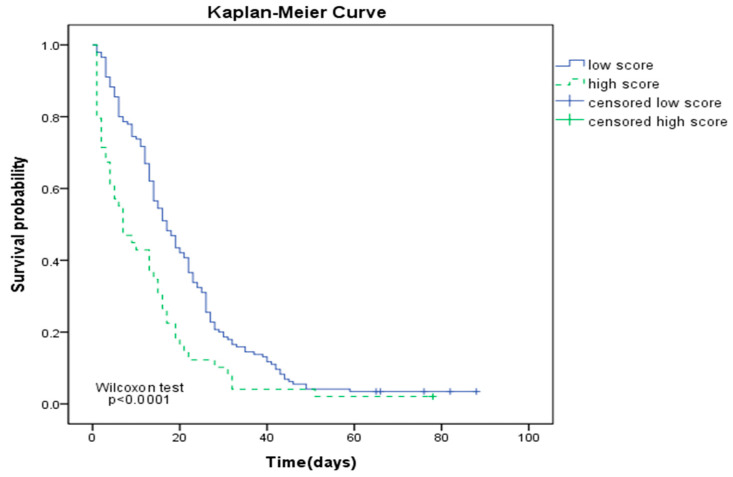
Kaplan–Meier curve of 60-day mortality for advanced cancer patients with SI > 0.94 (high score) and SI < 0.94 (low score).

**Table 1 ijerph-17-04904-t001:** Comparison of the medical history of patients, survivors versus nonsurvivors at 60 days after the index emergency department visit.

Variable	Patients			*p*-Value
	Total	Survivors	Nonsurvivors	
No.	305	117	188	
Age	63.50 ± 13.29	65.10 ± 12.42	62.51 ± 13.74	0.0971
Male	195 (63.93)	71 (60.68)	124 (65.96)	0.4179
Primary cancer				
Thyroid cancer	2 (0.66)	2 (1.71)		0.1464
Hypopharyngeal cancer	13 (4.26)	8 (6.84)	5 (2.66)	0.0885
Lung cancer	86 (28.20)	34 (29.06)	52 (27.66)	0.8939
Oropharyngeal cancer	10 (3.28)	3 (2.56)	7 (3.72)	0.7464
Nasopharyngeal cancer	8 (2.62)	5 (4.27)	3 (1.60)	0.2675
Oesophageal cancer	17 (5.57)	5 (4.27)	12 (6.38)	0.6001
Gastric cancer	29 (9.51)	9 (7.69)	20 (10.64)	0.5143
Colon cancer	43 (14.10)	17 (14.53)	26 (13.83)	0.9987
Rectal cancer	16 (5.25)	5 (4.27)	11 (5.85)	0.7363
Bladder cancer	4 (1.31)	1 (0.85)	3 (1.60)	0.9700
Renal cancer	6 (1.97)	2 (1.71)	4 (2.13)	0.7981
Prostate cancer	7 (2.30)	3 (2.56)	4 (2.13)	0.8045
Cervical cancer	4 (1.31)	1 (0.85)	3 (1.60)	0.9716
Uterine cancer	3 (0.98)		3 (1.60)	0.2883
Ovarian cancer	3 (0.98)	1 (0.85)	2 (1.06)	0.8500
Brain cancer	2 (0.66)	2 (1.71)		0.1464
Pancreatic cancer	12 (3.93)	4 (3.42)	8 (4.26)	0.9501
Hepatic cell cancer	21 (6.89)	7 (5.98)	14 (7.45)	0.7961
Cholangial cancer	8 (2.62)	4 (3.42)	4 (2.13)	0.4883
Breast cancer	17 (5.57)	7 (5.98)	10 (5.32)	0.8059
Soft tissue cancer	6 (1.97)		6 (3.19)	0.0857
Previous treatment				
Chemotherapy	255 (83.61)	98 (83.76)	157 (83.51)	0.9543
Radiotherapy	157 (51.48)	57 (48.72)	100 (53.19)	0.5207
Target therapy	90 (29.51)	37 (31.62)	53 (28.19)	0.6100
Surgical treatment	158 (51.80)	61 (52.14)	97 (51.60)	0.9267
Comorbidities				
Diabetes mellitus	69 (22.62)	25 (21.37)	44 (23.40)	0.7851
Hypertension	97 (31.80)	44 (37.61)	53 (28.19)	0.1117
Cerebrovascular accident	9 (2.95)	4 (3.42)	5 (2.66)	0.9736
Heart failure	5 (1.64)	1 (0.85)	4 (2.13)	0.6525
Coronary artery disease	14 (4.59)	4 (3.42)	10 (5.32)	0.6243
Chronic obstructive pulmonary disease	16 (5.25)	3 (2.56)	13 (6.91)	0.1636
End stage renal disease	6 (1.97)	2 (1.71)	4 (2.13)	0.7981
Liver cirrhosis	25 (8.20)	7 (5.98)	18 (9.57)	0.3696

**Table 2 ijerph-17-04904-t002:** Comparison of the clinical findings of patients, survivors versus nonsurvivors at 60 days after the index emergency department visit.

Variable	Patient							
	Total	Survivors	Nonsurvivors	*p*-Value	Univariate OR (95% CI)		Multiple OR ** (95% CI)	
No.	305	117	188					
Body temperature (℃) *	36.77 ± 1.32	37.11 ± 1.33	36.6 ± 1.27	0.0006	0.73	(0.61, 0.87)		
Pulse rate (/min) *	110.31 ± 21.17	102.40 ± 19.55	115.30 ± 20.67	<0.0001	1.03	(1.02, 1.05)		
Respiratory rate (/min) *	22.53 ± 5.49	21.10 ± 4.30	23.41 ± 5.96	0.0006	1.1	(1.04, 1.16)		
Systolic blood pressure (mmHg) *	104.82 ± 20.91	114.50 ± 22.01	98.81 ± 17.77	<0.0001	0.96	(0.95, 0.97)		
Mean arterial pressure (mmHg) *	77.57 ± 16.13	83.61 ± 16.31	73.82 ± 14.87	<0.0001	0.96	(0.95, 0.98)		
Shock index *	1.11 ± 0.35	0.92 ± 0.27	1.21 ± 0.35	<0.0001	1.37	(1.24, 1.51)	1.39	(1.24, 1.55)

* indicates statistical significance; ** performed by logistic regression model adjusted for age, gender, personal medical and medication history

**Table 3 ijerph-17-04904-t003:** Optimal cut-off value for the SI with corresponding accuracy, sensitivity, and specificity.

Optimal Cut-Off	Accuracy Rate	Sen	Sp	PPV	NPV	False +ve	False −ve
0.94	73.11%	81.38%	59.83%	76.50%	66.67%	15.26%	11.54%

Sen—sensitivity; Sp—specificity; PPV—positive predictive value; NPV—negative predictive value; +ve—positive; −ve—negative.

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
