# Peer review of "Shock Index: A Simple and Effective Clinical Adjunct in Predicting 60-Day Mortality in Advanced Cancer Patients at the Emergency Department"

_ijerph, 2020, doi:10.3390/ijerph17134904_

Round 1

Reviewer 1 Report

Dear Authors

The study described in the manuscript is quite interesting, and Shock Index really might be no-cost adjunct to swift assessment of advanced cancer patients not only in Emergency Departments. Clinicians do need swift and easy tools for screening such a group of patients (as for example lately published in Journal of Palliative Care study on Suprise Question and ECOG score)

The study design seems correct to me, the number of recruited participants is powerfull enough. The statistical methods used in the study are correct and well described.

I would just think to compute and add the Harrell's C-index to be sure that the scores are good in determining survivors and non-survivors- how would you comment on this?

As a results are well discussed, I still lack the more information and ephasis on limitations (eg how the concurent diseases and medications with potential influence on vital signs might biased the results- was such data collected at all?)- could you respond?

Summarising- I think that the manuscript is worth publishing and the described tool is of value not only for ED clinicians but more widely (even for Palliative Care Deps at first admssion visits).

Author Response

Response to reviewer #1

#1: I would just think to compute and add the Harrell's C-index to be sure that the scores are good in determining survivors and non-survivors- how would you comment on this?

Response #1:

The Harrell's C-index is 0.74. We have added the result of Harrell's C-index at line 167-168 (Results section, page 5):

Multiple logistic regression analysis adjusted for age, sex, personal medical and medication history found that the odds of death at 60 days after the index visit was 1.39 (95% CI: 1.24-1.55; p<0.0001) times higher among participants with each 0.1 increment of the index. Area under ROC curve was found to be 0.7511 (Hosmer-Lemeshow statistic p-value 0.2878) (Figure 1). The Harrell's C-index is 0.74.

#2: As a results are well discussed, I still lack the more information and ephasis on limitations (eg how the concurent diseases and medications with potential influence on vital signs might biased the results- was such data collected at all?)- could you respond?

Response #2:

We have gone through our records to ensure that none of our subjects are using antiarrhythmic agent or received pacemaker implantation. Some patients are on hypertensive medications. We have acknowledged that this may be a limitation of our study. However, we have performed logistic regression model adjusted for patients’ concurrent diseases and medications with potential influence on vital signs.

We have added the following paragraph in our Discussion section, last paragraph (page 8, line 238-240):

Although our study model were adjusted accordingly to patients' medications and concurrent diseases during analysis, future usage of shock index may need to take these into account which may affect vital sign readings.” The results of this study should be prospectively validated with a larger sample population of advanced cancer patients, and ideally with different subsets of patients with different advanced cancers. This would help to ensure its replicability across all types of cancer. Further studies into other scoring systems suitable for the ED setting may also be considered.

Reviewer 2 Report

Cheng TH et al., investigated importance of shock index (SI) to predict 60 day mortality in patient with advanced cancer the emergency department. They reported SI is important factor to predict it. They said SI is an ideal evaluation tool to rapidly predict short to middle mortality. This reviewer has some comments on this manuscript.

1. The authors showed results of univariate analysis for predicting 60-day mortality. However, results of multivariate analysis also should be shown after the results of univariate analysis in order to exclude cofounding factor and confirm that SI is one of independent predictors. 

2. Univariate analysis showed that there are other important factors which can be rapidly obtained at emergent department. If BMI or respiratory rate (RR) are also significant factor in multivariate analysis, please provide cut-off value of them to predict mortality.

3. Also, when combining these factors (SI, BMS, RR), better scale or scoring system can be provided. Therefore, the author should provide scoring system combining these independent predictors to predict mortality. Afterwards, the rate of mortaliy using Kaplan-Maier curve between each score should be provided to understand how it works. 

Author Response

Response to reviewer #2

#1: The authors showed results of univariate analysis for predicting 60-day mortality. However, results of multivariate analysis also should be shown after the results of univariate analysis in order to exclude cofounding factor and confirm that SI is one of independent predictors.

Response #1:

We have performed multivariate analysis after the results of univariate analysis and confirmed that SI is the only independent predictor (Results section, page 5-6, line 164-172). Result of multivariate analysis has also been added in Table 2 (Results section, page 5).

Table 2. Comparison of clinical findings of patients, survivors versus non-survivors at 60 days after index emergency department visit.

Variable

Patient

Total

Survivors

Non-survivors

p-value

Univariate
OR (95%CI)

Multiple
OR** (95%CI)

No.

305

117

188

Body temperature (℃) *

36.77 ± 1.32

37.11 ± 1.33

36.6 ± 1.27

0.0006

0.73

(0.61, 0.87)

Pulse rate (/min) *

110.31 ± 21.17

102.40 ± 19.55

115.30 ± 20.67

<.0001

1.03

(1.02, 1.05)

Respiratory rate (/min) *

22.53 ± 5.49

21.10 ± 4.30

23.41 ± 5.96

0.0006

1.1

(1.04, 1.16)

Systolic blood pressure (mmHg) *

104.82 ± 20.91

114.50 ± 22.01

98.81 ± 17.77

<.0001

0.96

(0.95, 0.97)

Mean arterial pressure (mmHg) *

77.57 ± 16.13

83.61 ± 16.31

73.82 ± 14.87

<.0001

0.96

(0.95, 0.98)

Shock index *

1.11 ± 0.35

0.92 ± 0.27

1.21 ± 0.35

<.0001

1.37

(1.24, 1.51)

1.39

(1.24, 1.55)

* indicates statistical significance.

**performed by logistic regression model adjusted for age, sex, personal medical and medication history

Multiple logistic regression analysis adjusted for age, sex, personal medical and medication history found that the odds of death at 60 days after the index visit was 1.39 (95% CI: 1.24-1.55; p<0.0001) times higher among participants with each 0.1 increment of the index. Area under ROC curve was found to be 0.7511 (Hosmer-Lemeshow statistic p-value 0.2878) (Figure 1). The Harrell's C-index is 0.74.

In multivariate logistic regression analysis, we found that SI was the only factor which was significantly related to short-term survival.

#2: Univariate analysis showed that there are other important factors which can be rapidly obtained at emergent department. If BMI or respiratory rate (RR) are also significant factor in multivariate analysis, please provide cut-off value of them to predict mortality.

Response #2:

In multivariate logistic regression analysis, we found that SI was the only factor which was significantly related to short-term survival. We confirmed that other factors are NOT significant in multivariate analysis. (Table 2) (Results section, page 5).

We have added “In multivariate logistic regression analysis, we found that SI was the only factor which was significantly related to short-term survival.” (Results section, page 6, line 171-172).

#3:Also, when combining these factors (SI, BMS, RR), better scale or scoring system can be provided. Therefore, the author should provide scoring system combining these independent predictors to predict mortality. Afterwards, the rate of mortaliy using Kaplan-Maier curve between each score should be provided to understand how it works.

Ans: As SI was the only factor which was significantly related to short-term survival, Kaplan–Meier method was performed to examine survival between groups with high versus low SI.

We have added the following description and Figure 2 for clarification:

Kaplan–Meier method was also performed to examine survival between groups with high versus low SI. (Materials and Methods: Statistical analysis section, page 3, line 141-142)

2.4. Statistical analysis

The means ± SD of numerical variables and frequencies with corresponding percentages of categorical variables were analyzed with univariate analyses of independent sample t-test and χ2 test respectively. Univariate and multivariate logistic regression analyses were subsequently employed to evaluate the odds ratio of clinical parameters and each SI increment of 0.1 with respect to 60-day mortality. A receiver operating characteristic (ROC) curve was also plotted to examine the predictive performance of SI, with optimal cut-off point identified via maximising Youden’s index. Kaplan–Meier method was also performed to examine survival between groups with high versus low SI. Statistical significance was taken at p < 0.05.

[Results section, page 6, line 176-181]:

Kaplan-Meier curve revealed that the 60-day mortality in advanced cancer patients with SI >0.94 were significantly higher than those with lower SI (Wilcoxon test p<0.0001) (Figure 2).

(Figure 2): Please refer to the attached document for Figure 2. Thanks

Figure 2. Kaplan-Meier curve of 60-day mortality for advanced cancer patients with SI >0.94 (high score) and SI <0.94 (low score).

Round 2

Reviewer 2 Report

I do not have any further comments.

Author Response

Response to reviewer #2

Response: We have asked a native English speaker to review our manuscript as requested. In the event you are of the opinion that it requires further editing, kindly let us know and please grant us a deadline extension so that we may send it to a language editing service as you have suggested.
